# COSP-RTTOV-1.0: Flexible radiation diagnostics to enable new science applications in model evaluation, climate change detection, and satellite mission design

Jonah K. Shaw[1,2], Dustin J. Swales[3], Sergio DeSouza-Machado[4], David D. Turner[3], Jennifer E. Kay[1,2], and David P. Schneider[2]

[1]Department of Atmospheric and Oceanic Sciences, University of Colorado Boulder, Boulder, Colorado
[2]Cooperative Institute for Research in Environmental Science, University of Colorado Boulder, Boulder, Colorado
[3]NOAA Global Systems Laboratory, Boulder, Colorado
[4]Joint Center for Earth Systems Technology/Physics Department, University of Maryland, Baltimore County, Baltimore, MD, USA

**Correspondence:** Jonah K. Shaw (jonah.shaw@colorado.edu)

**Abstract.** Infrared spectral radiation fields observed by satellites make up an information-rich, multi-decade record with continuous coverage of the entire planet. As direct observations, spectral radiation fields are also largely free from uncertainties that accumulate during geophysical retrieval and data assimilation processes. Comparing these direct observations with earth system models (ESMs), however, is hindered by definitional differences between the radiation fields satellites observe and those generated by models. Here, we present a flexible, computationally efficient tool called COSP-RTTOV for simulating satellite-like radiation fields within ESMs. Outputs from COSP-RTTOV are consistent with instrument spectral response functions and orbit sampling, as well as the physics of the host model. After validating COSP-RTTOV's performance, we demonstrate new constraints on model performance enabled by COSP-RTTOV. We show additional applications in climate change detection using the NASA AIRS instrument, and observing system simulation experiments using the NASA PREFIRE mission. In summary, COSP-RTTOV is a convenient tool for directly comparing satellite radiation observations with ESMs. It enables a wide range of scientific applications, especially when users desire to avoid the assumptions and uncertainties inherent in satellite-based retrievals of geophysical variables or in atmospheric reanalysis.

## 1 Introduction

Comparisons between models and satellite observations enable science that combines the predictive power of models with the real-world constraint of observations (Simpson et al., 2025). Climate models represent our theoretical understanding of how the climate operates. Models of varying complexity offer testbeds to study the physics and boundary conditions necessary to reproduce observed phenomena. When appropriately validated, models provide immense societal benefit: Regional forecast models predict local weather, and fully-coupled earth system models (ESMs) project decadal and centennial climate changes to inform policy and mitigation efforts (Eyring et al., 2021; Lee et al., 2021). While models offer the flexibility to look forward and replay time with different physics, satellites observe the "true" state of the climate system. Satellites observe the evolution

of the atmospheric state that models aim to reproduce, providing constraints and test cases for models. Additionally, long satellite records document climate change and allow for its attribution to human actions (e.g. Raghuraman et al., 2021). When appropriately compared, models and observations can answer scientific questions that are not tractable for either tool alone.

Critically, the powerful synergistic uses of models and observations require consistent definitions of climate variables. In models, the state of the surface and atmosphere is contained in profiles of geophysical variables such as temperature, humidity, and trace gas concentrations. These geophysical variables are taken to be representative of mean values for a single model gridcell, which may range in size from hundreds of meters to hundreds of kilometers. Passive satellite observations, on the other hand, only measure spectrally-resolved radiation fields at the top of the atmosphere. These spectral radiation fields are commonly measured in units of radiance or brightness temperature. Spectral radiation fields depend on the specific optics of observing instruments, and often have spatial footprints much smaller than ESM gridcells. Furthermore, polar-orbiting satellites only view the earth's surface at certain times of day, while ESMs update geophysical variables at every model timestep. Useful comparisons between models and observations must reconcile these inconsistencies in how each set of data is produced.

The need for common climate variables for model-satellite comparisons has led to the development of satellite simulators. In brief, satellite simulators operate by first simulating the radiation fields observed by a satellite using model fields and then emulating the process of inferring the underlying geophysical state from those fields (satellite retrievals). This two-step process produces "satellite-like" fields to compare with observations. Satellite simulators have been used to study the earth's climate with both active and passive satellite observations for more than two decades (e.g. Klein and Jakob, 1999; Webb et al., 2001; Zhang et al., 2005; Williams and Tselioudis, 2007). While satellite simulators have their own limitations (e.g. Pincus et al., 2012; Mace et al., 2011), comparisons with observations are vastly improved over less-nuanced methods. An important benefit of satellite simulators is that comparisons are performed with geophysical variables and can be easily related to climate processes. As a result, most ESM-observation comparisons use geophysical variables. Such comparisons are also convenient because global, gridded geophysical variables are widely available from reanalysis products and Level 3 and 4 satellite retrievals. However, geophysical variables produced using satellite retrievals or data assimilation also have epistemic uncertainty resulting from inferring a geophysical variable (e.g. humidity) from an observed field (e.g. radiance). This epistemic uncertainty is rarely quantified or reported in global datasets used to evaluate ESMs. Radiation fields, on the other hand, are derived directly from L0 observations and can be accurately calculated in ESMs using radiative transfer models. This makes radiation fields more certain than geophysical variables but less interpretable. These respective advantages and limitations determine the appropriate scientific applications model-observation comparisons using radiation fields and geophysical variables.

One satellite simulator program commonly used for geophysical variable comparisons is the Cloud Feedback Model Intercomparison Project (CFMIP) Observation Simulator Package (COSP) (Bodas-Salcedo et al., 2011; Swales et al., 2018). COSP combines simulators for multiple platforms in a single open-source software package. It also allows for subgrid sampling of ESM fields to address differences in the spatial scales of models and observations. The satellite-like cloud fields produced by COSP have been widely used to evaluate the representation of clouds in ESMs in model intercomparison projects (e.g., Nam et al., 2012; Bodas-Salcedo et al., 2014), between successive model generations (e.g., Kay et al., 2012; Medeiros et al., 2023), and in studies

of specific regions and climate processes (e.g., Kay et al., 2016; Shaw et al., 2022). As a whole, COSP enables understanding of how clouds and their radiative effects respond to different model parameterizations, choices of tuning parameters, and the changing climate.

While geophysical variable comparisons are well-suited to the evaluation and comparison of model processes, changes in the optical, or the radiative representation of the atmosphere are more appropriate for climate change detection studies, constraints on aggregate model behavior, and satellite mission design. In short, climate change detection involves distinguishing an observed signal from internal climate variability. Detection studies benefit from the known uncertainty in satellite radiance fields (e.g. Strow and DeSouza-Machado, 2020), which in some cases may be tied to absolute radiometric standards (Feldman et al., 2011a; Wielicki et al., 2013; Huang et al., 2022). As a constraint on models, radiation comparisons can expose compensating biases hidden in broadband radiation fields (e.g., Huang et al., 2007). Another exciting application of simulating climate-scale radiation fields is for Observing System Simulation Experiments (OSSEs) (Feldman et al., 2011b, 2015). OSSEs simulate the observations of satellite platforms in order to assess their scientific value, but are rarely run at climate timescales in coupled models (Feldman et al., 2011b, 2015; Hoffman and Atlas, 2016; Zeng et al., 2020). Conducting climate OSSEs during the design and evaluation of proposed missions can identify weaknesses and improvement areas before satellites are built and launched.

Despite these applications, radiation-based comparisons are not commonly used. One reason is their technical complexity. Simulating satellite radiances requires feeding the instantaneous state of the surface and atmosphere into a separate radiative transfer model. This step requires saving large volumes of model output. Users must also simulate radiation fields tailored to how specific satellite instruments and channels respond to radiation at different wavelengths (instrument spectral response functions). A radiative transfer tool that runs inline with model physics and directly simulates instrument radiances would remove these technical hurdles and democratize the advantages of radiation-based model-observation comparisons.

Here, we present a flexible and computationally efficient tool for the simulation of spectral radiation fields within ESMs. By coupling the Radiative Transfer for TOVS (RTTOV) radiative transfer model (Saunders et al., 2018) into COSP, this tool (COSP-RTTOV) combines the diverse uses of a highly-developed radiative transfer model with the user community and practicality of a popular satellite simulator package (COSP). COSP-RTTOV can simulate radiation fields in both cloudy and clear-sky atmospheres. Output radiation fields are specific to individual instrument platforms and users may specify the viewing and orbital geometries. We additionally extend the implementation of flexible satellite-like sampling patterns in COSP-RTTOV to all simulator fields available in COSP. In this paper, we begin by describing the implementation and design of this tool. We then run COSP-RTTOV in single-column and global configurations of an ESM to validate its performance and estimate the computational cost. Finally, we demonstrate applications of COSP-RTTOV in climate change detection, model evaluation, and satellite mission design.

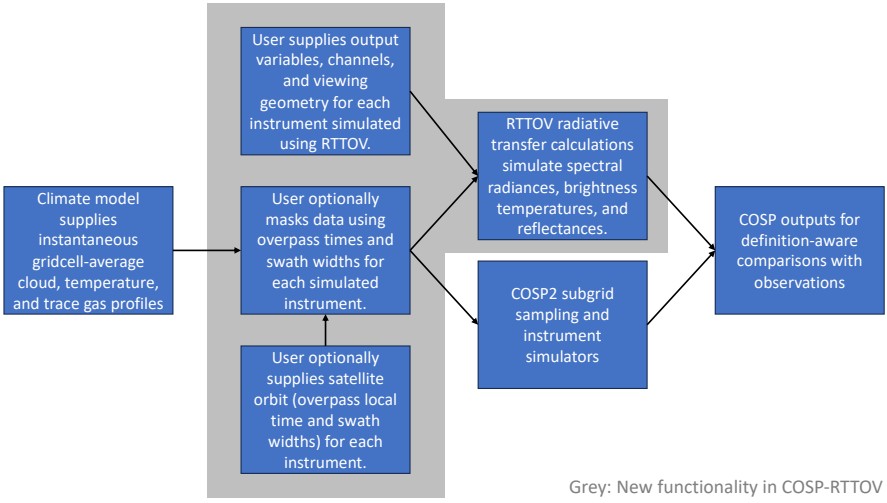

**Figure 1.** COSP-RTTOV Schematic.

## 2 Methods

COSP-RTTOV adds the flexible simulation of spectral radiation fields to the functionality of COSP2 (Swales et al., 2018). As a secondary improvement, we also enable simple satellite-like sampling patterns for COSP2 retrievals. The simulation of spectrally-resolved radiation fields is accomplished by coupling the RTTOV v13 radiative transfer model (Saunders et al., 2018) with COSP2. The addition of satellite-like sampling patterns is accomplished by limiting simulator calculations to gridcells that fall within user-specified viewing swaths. Figure 1 shows a simplified flowchart of COSP-RTTOV operations, with new functionality relative to COSP2 boxed in grey.

### 2.1 Flexible Simulation of Spectral Radiation Fields

#### 2.1.1 Coupling with RTTOV Radiative Transfer Model

We use the RTTOV radiative transfer model to efficiently generate spectral radiation fields over the entire globe for century-scale ESM simulations. RTTOV is a fast radiative transfer model for simulating satellite radiances. RTTOV can account for the different spectral response functions of spectral channels on various instrument platforms, which is essential for enabling consistent comparisons with observations. For complete RTTOV documentation we refer readers to Saunders et al. (2018).

To produce spectral radiation fields, gridcell-average surface properties and profiles of temperature, trace gas concentrations, and cloud properties are passed from an ESM into RTTOV via COSP. To determine what radiation fields are produced, users specify which subset of spectral channels and output fields should be produced for each instrument being simulated by RTTOV. All user specifications available in COSP-RTTOV are summarized in Table 1.

**Table 1.** User specifications instruments simulated in COSP-RTTOV.

| Category | User Inputs |
|---|---|
| Requested Outputs | Compute radiances? |
| | Compute brightness temperatures? (LW only) |
| | Compute all-sky and cloudy fields? |
| Radiative Transfer Specifications | RTTOV trace gas coefficients file path |
| | RTTOV cloud coefficients file path |
| | Include $SO_2$ in radiative transfer? |
| | Include $N_2O$ in radiative transfer? |
| | Include $CO$ in radiative transfer? |
| | Include $CO_2$ in radiative transfer? |
| | Include $CH_4$ in radiative transfer? |
| | Include $O_3$ in radiative transfer? |
| If using uniform fixed trace gas concentrations instead of interactive model trace gas fields | $SO_2$ mixing ratio |
| | $N_2O$ mixing ratio |
| | $CO$ mixing ratio |
| | $CO_2$ mixing ratio |
| | $CH_4$ mixing ratio |
| | $O_3$ mixing ratio |
| For hyperspectral sounders | Use Principal Component RTTOV (PC-RTTOV)? |
| | Number of PC-RTTOV predictors |
| | Number of PC-RTTOV principal components |
| If using satellite-like sampling patterns | Number of sampling patterns |
| | Local time for the center of each sampling pattern |
| | Width of each sampling pattern (km) |

### 2.1.2 Earth System Model Experiments with CESM2 and COSP-RTTOV

We run multiple model experiments using the Community Earth System Model Version 2 (CESM2) (Danabasoglu et al., 2020) as the host model for COSP-RTTOV. We use the single-column version of CESM2's atmospheric component (Gettelman et al., 2019) to validate the performance of COSP-RTTOV. We additionally use global atmosphere-only simulations to demonstrate the applications of COSP-RTTOV. All experiments use the CESM2.1.5 release.

We run single-column experiments for 7 Intensive Observation Period (IOP) cases that sample a broad variety of atmospheric 110 and cloud conditions (see Gettelman et al. (2019) Table 1 reproduced here as Table 2). IOPs range from 17 to 30 days and all radiation fields are computed hourly. In these experiments, all-sky spectral irradiance fields are produced from CESM2's internal radiative transfer scheme for comparison with COSP-RTTOV (Section 2.1.4). CESM2 uses the Rapid Radiative

**Table 2.** Single-column model intensive observation periods. Reproduced from Gettelman et al. (2019).

| Name | Long Name | Lat | Lon | Date | Days | Reference | Type |
|------|-----------|-----|-----|------|------|-----------|------|
| arm97 | ARM Southern Great Plains | 36 | 263 | Jun. 1997 | 30 | Zhang et al. (2016) | Land convection |
| cgilsS6 | CFMIP-GASS SCM/LES Intercomp | 17 | 211 | Jul. 1997 | 17 | Zhang et al. (2013) | Shallow cumulus |
| cgilsS11 | CFMIP-GASS SCM/LES Intercomp | 32 | 231 | Jul. 1997 | 32 | Zhang et al. (2013) | Stratocumulus |
| cgilsS12 | CFMIP-GASS SCM/LES Intercomp | 35 | 235 | Jul. 1997 | 35 | Zhang et al. (2013) | Stratus |
| twp06 | Tropical W. Pacific Convection | -12 | 131 | Jan. 2006 | 26 | May et al. (2008) | Tropical convection |
| mpace | Mixed Phase Arctic Clouds Exp | 71 | 206 | Oct. 2004 | 17 | Verlinde et al. (2007) | Arctic |
| sparticus | Small Particles in Cirrus | 37 | 263 | Apr. 2010 | 30 | Mace et al. (2009) | Cirrus, convection |

Transfer Model longwave (RRTMG-LW) radiation scheme (Mlawer et al., 1997; Pincus et al., 2003), which divides longwave radiation into 16 spectral bands from 10-3250 $\mathrm{cm}^{-1}$. When evaluating the clear-sky radiation fields produced by COSP-RTTOV, we use the arm97 IOP (Section 2.1.3).

We run two global atmosphere-only experiments to compare with the observational satellite record and quantify internal climate variability. In both experiments, only a subset of AIRS channels are simulated to limit computational cost and the volume of data produced. The first experiment runs from 2000-2022 using observed sea surface temperatures and sea ice fields as boundary conditions. Atmospheric forcings are taken from the CMIP6 AMIP protocol from 2000-2014 and from the CMIP6 SSP3-7.0 scenario from 2015-2022. This experiment is intended to be compared with observations. The second experiment uses surface boundary conditions and atmospheric forcings from the CESM2 pre-industrial control experiment. Specifically, sea surface temperature and sea ice fields are taken from years 501-699 of the pre-industrial experiment. This experiment quantifies internal variability in an unforced pre-industrial climate for use in climate change detection studies.

### 2.1.3 Validation of clear-sky brightness temperatures against SARTA

The primary function of COSP-RTTOV is to produce synthetic radiance and brightness temperature fields that are consistent with satellite observations. We accomplish this by comparing COSP-RTTOV to a radiative transfer tool used by NASA's Atmospheric Infra-Red Sounder (AIRS) mission. The Stand-alone AIRS Radiative Transfer Algorithm (SARTA) (Strow et al., 2003, 2006; Desouza-Machado et al., 2020) is a fast radiative transfer model for simulating AIRS radiances, given realistic atmospheric profiles of water vapor, ozone, and temperature, as well as other parameters such as surface emissivity.

SARTA's accuracy and wide spectral coverage make it an excellent validation tool for COSP-RTTOV. We specifically compare against 2645 AIRS L1C channels. These spectral channels are corrected for drift and shifted to a fixed frequency grid, making them ideal for long-term comparisons. Figure 2 compares clear-sky brightness temperatures produced by COSP-RTTOV and SARTA for the arm97 IOP (Table 2). Overall, Figure 2 demonstrates that COSP-RTTOV simulates accurate clear-sky radiation fields for comparisons with infrared sounders. Good agreement (mean error < 1 K, error standard deviation <0.5 K) is shown for virtually all spectral regions. Larger differences in the 667-668$\mathrm{cm}^{-1}$ spectral region result from different assumptions of the atmospheric column above the CESM2 model top between COSP-RTTOV and SARTA. Thus discrepancies

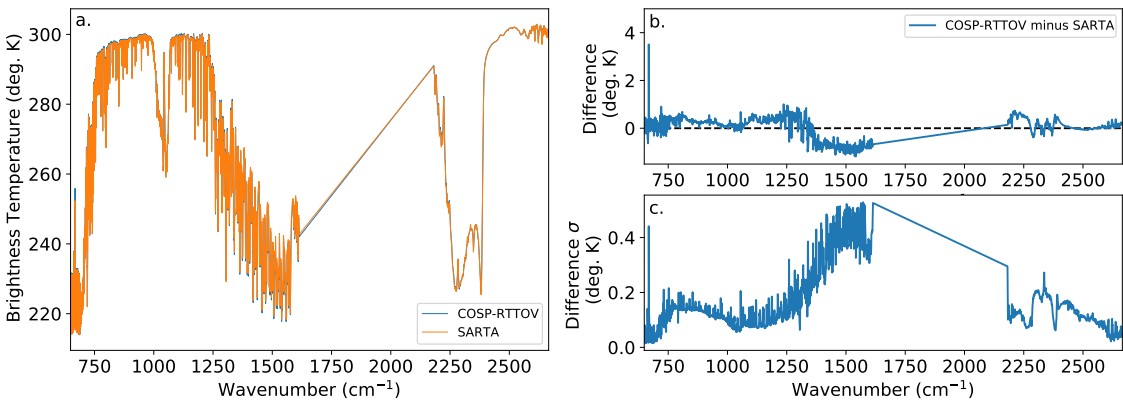

**Figure 2.** Comparison of brightness temperatures produced by COSP-RTTOV and SARTA for AIRS L1C channels. a. Simulated brightness temperatures across the AIRS spectral region ($3.7 - 15.4\mu m$). b. Mean and c. Standard deviation of COSP-RTTOV brightness temperature differences relative to SARTA. Brightness temperatures are computed for 333 atmospheric profiles taken from a single-column mid-latitude simulation of COSP-RTTOV (see section 2.1.4).

here are not relevant to the intended applications of COSP-RTTOV. Because COSP-RTTOV is intended for direct radiation comparisons and not geophysical retrievals, a more detailed accounting of brightness temperature differences is not needed.

### 2.1.4 Validation of all-sky irradiance against CESM2 and RRTMG

We next validate COSP-RTTOV against the all-sky radiation fields produced by CESM2. A key goal of COSP2 was to allow for greater consistency between the cloud properties used in the host model and those used to produce COSP's diagnostic outputs. RTTOV, however, has its own schemes for cloud optics and cloud overlap assumptions (how clouds at different vertical levels are distributed at sub-grid scales). Specifically, users may select cloud optics from multiple parameterizations, and choose between maximum/random and simplified two-column cloud overlap schemes (Saunders et al., 2018). Effectively, this means

that the speed of RTTOV calculations comes at the expense of not being able to ensure consistency with the host model.

To understand the influence of different radiative transfer assumptions on the radiative properties of clouds, we compare COSP-RTTOV with RRTMG-LW. To compare against spectral irradiance fields from RRTMG-LW, we produce RRTMG-like all-sky irradiances from COSP-RTTOV. These RRTMG-like irradiances are computed by simulating radiance fields at high spectral resolution for multiple viewing angles, summing over the appropriate spectral interval, and performing a quadrature

over solid angle. Specifically, we use channels based on the spectral response functions of two Fourier Transform Spectrometer instruments, the Infrared Atmospheric Sounding Interferometer (IASI) and the Far-infrared Outgoing Radiation Understanding and Monitoring (FORUM) mission's interferometer. We simulate IASI-like channels to cover the $700 - 2600 \text{cm}^{-1}$ region and FORUM-like channels to cover the $100 - 700 \text{cm}^{-1}$ region. The IASI (FORUM) channels have $0.25(0.3)\text{cm}^{-1}$ spacing and $0.5(0.5)\text{cm}^{-1}$ resolution (FWHM). This spectral range allows us to compare against 14 of 16 RRTMG-LW channels.

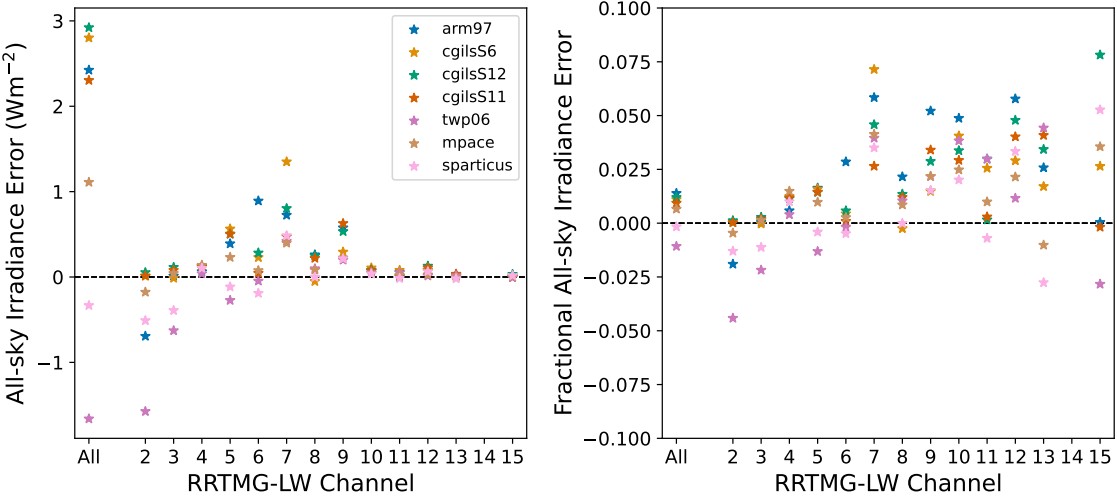

**Figure 3.** Comparison of all-sky irradiances produced by COSP-RTTOV and CESM2 for 14 spectral bands in RRTMG-LW. a. Mean all-sky irradiance error. b. Fractional all-sky irradiance error. Temporal averages are taken before comparison between COSP-RTTOV and RRTMG-LW. Spectral boundaries in cm$^{-1}$ for RRTMG-LW bands 2-15 are 350, 500, 630, 700, 820, 980, 1080, 1180, 1390, 1480, 1800, 2080, 2250, 2380, 2600 (e.g. Band 3 spans 500-630 cm$^{-1}$). Band 14 samples the mesosphere above the CESM2 model top and is excluded from this analysis. RTTOV radiances are converted to irradiances using a 6-point gaussian quadrature with viewing zenith angles and weights following Stamnes et al. (2017).

Comparing COSP-RTTOV against the RRTMG spectral irradiances requires simulating tens of thousands of individual spectral channels at each time step for a single atmospheric column. Computing and saving this many additional output fields for every grid cell of an ESM is not computationally feasible, so we validate using single-column experiments. The single-column IOP cases (Table 2) allow us to confirm that COSP-RTTOV fields consistently represent the cloud properties of CESM2.

Figure 3 compares the RRTMG-LW and RRTMG-like all-sky irradiances for all single-column IOPs. Because COSP-RTTOV and RRTMG-LW see identical model states, differences between them result only from the models themselves and the conversion of COSP-RTTOV radiances to RRTMG-like irradiances. The total all-sky irradiance error summed across all bands never exceeds 3 Wm$^{-2}$ and the total fractional error never exceeds 2%. Fractional errors for individual spectral bands are larger, but never exceed 10% and are almost always less than 5%. Because regional all-sky model biases are often greater than

these values, we find this level of agreement to be appropriate for the intended applications of COSP-RTTOV. Furthermore, the presence of similar irradiance biases in an analogous clear-sky comparison (Figure A1) despite strong agreement in clear-sky radiances with SARTA (Figure A2) indicates that the radiance to irradiance conversion is likely the main source of error. Because COSP-RTTOV is intended for brightness temperature and radiance comparisons, we are thus confident that Figure 3 indicates a conservative upper bound on all-sky errors. Overall, Figures 2 and 3 demonstrate that COSP-RTTOV appropriately

simulates satellite radiances and replicates the cloud properties of the host model.

**Table 3.** Computational cost of COSP-RTTOV.

| Experiment Name | COSP2 simulators | COSP-RTTOV Outputs | Total Computation Cost (pe-hrs/simulated year)[1] | % cost increase relative to $CAM\_only$ |
|---|---|---|---|---|
| $CAM\_only$ | None. | N/a | 1830 | N/a |
| $CAM\_RTTOV$ | 45 AIRS Channels[2] | Radiance and BT | 2755 | 51% |
| $CAM\_RTTOV_{swathed}$[3] | 45 AIRS Channels[2] | Radiance and BT | 2210 | 21% |

[1]CESM2 is run at 1-degree resolution in an atmosphere-only configuration using eight nodes on the NCAR Derecho system. [2]Radiances fields are produced as clear, cloudy, and total sky averages. Brightness temperatures are produced as clear and total sky averages. [3]Orbit centered at 1:30pm local time with 1800km swath width.

## 2.2 Satellite-like Sampling Patterns

To specify satellite-like sampling patterns, users may supply a list of sampling local times (in units of hours) and swath widths (in units of kilometers). The sampling "local time" refers to a linear shift from UTC as a function of a gridcell's longitude ($t_{local} = t_{UTC} - longitude * 24/360$). This specification mimics satellite overpass times and ensures consistent sampling of the diurnal cycle. Conversely, the "swath width" determines the spatial region around each local time that is simulated. Supplying a swath width in units of distance rather than radians produces a larger sampling density at higher latitudes that is consistent with observations. By specifying any number of sampling local times and swath widths, users can emulate output comparable to a single daytime or nighttime instrument or simulate an entire constellation of identical instruments with different orbits. Applying these sampling patterns also reduces computational cost because simulators are run only on a subset of gridcells (see Section 2.3). If sampling patterns are not specified, outputs are computed for all gridcells at each timestep as in previous versions of COSP.

## 2.3 Computational Cost

To quantify the computational cost of running COSP-RTTOV, we report the cost of global atmosphere-only CESM2 experiments with different COSP-RTTOV configurations. Table 3 describes each experiment and reports the computational cost. Simulating spectral fields using RTTOV is more expensive than running the standard COSP2 simulators. However, applying reasonable swathing patterns cuts these costs noticeably.

## 3 Simulator Applications

Having described the simulator design and validation in Section 2, we now demonstrate the utility of global simulations of spectral radiation fields (Section 3.1) and satellite-like sampling patterns (Section 3.2).

## 3.1 Simulation of spectral radiation fields

### 3.1.1 Model Evaluation

Evaluating models against direct radiation observations gives insight into model biases without the epistemic uncertainty present in geophysical retrievals or renanalysis products. To be valuable for model developers, however, radiation fields should be intuitively related to geophysical variables and climate processes. Figure 4 shows global maps of simulated brightness temperatures paired with the corresponding geophysical fields that they effectively capture. Clear-sky radiation in transparent spectral regions effectively view surface temperatures (Figure 4a,b). Similarly, a channel sensitive to the upper-troposphere strongly resembles 274mb temperatures (Figure 4c,d). Finally, simple radiation comparisons can also capture cloud fields in regions with high thermal contrast (Figure 4e,f). Overall, Figure 4 demonstrates that appropriately chosen spectral channels enable ESM evaluation against direct observables without the loss of physical intuition often associated with radiation fields.

Once the relationships between radiation fields and geophysical variables have been established, we can easily evaluate model performance. Figure 5 compares COSP-RTTOV output with AIRS observations radiances from the same spectral channels shown in Figure 4. Results are shown for a tropical region in Equatorial Pacific (left column) and a land surface in the mid-latitudes (right column). This comparison identifies a wintertime cold bias at mid-latitudes (Figure 5b). The upper-troposphere channel (Figure 5c,d) shows both a cold bias and a secular cooling trend in agreement with the AIRS observations. Finally, the simple cloud amount metric in Figure 5e demonstrates that CESM2 captures the interannual variability from AIRS observations.

### 3.1.2 Climate Change Detection

In addition to providing a strict constraint on model performance, COSP-RTTOV also enables the use of radiation records in studies of climate change detection and attribution. Specifically, centennial-scale pre-industrial control simulations can characterize internal climate variability in radiation fields (e.g. Shaw and Kay, 2023). Distinguishing observed change from this internal variability enables the attribution of observed changes to anthropogenic forcing. Figure 6 compares observed AIRS radiance trends with internal variability generated from a pre-industrial control simulation run with COSP-RTTOV. Specifically, we examine the signal of CO2 increase in the upper-troposphere channel in Figure 5c,d. For both tropical and mid-latitude regions, AIRS detects a forced change within 15 years. The observed cooling trend is the result of increasing CO2 concentrations and upper-tropospheric temperature change. Increased CO2 concentrations raise thermal emission to higher and colder pressure levels, which themselves are also cooling. By producing satellite-like radiation fields from a long pre-industrial control simulation using COSP-RTTOV, this approach can be flexibly applied to multiple spectral channels and spatial regions.

### 3.1.3 Climate Observing System Simulation Experiments (OSSEs)

COSP-RTTOV allows us to easily run climate OSSE experiments for evaluating proposed missions and placing short satellite missions into the broader context of forced change and internal climate variability. Figure 7 shows one example using the

recently-launched NASA PREFIRE mission (L'Ecuyer et al., 2021). Climatological averages of PREFIRE-like radiances (Figure 7, top row) are computed by running COSP-RTTOV in historical ESM simulations. Additionally, annual time series from the same ESM experiment quantify forced change and internal variability (Figure 7, bottom row). While PREFIRE channels centered 12.4μm and 14.2μm respectively sample the atmospheric window and $CO_2$ absorption, additional channels at 20.6μm and 36.8μm sample features in the previously unobserved far-infrared. Evaluating PREFIRE observations against this synthetic record allows differences resulting from model physics to be separated from internal climate variability. Overall, producing PREFIRE-like radiances with COSP-RTTOV in long historical simulations provides a longer context in which to interpret a short observational record.

### 3.2 Satellite-like sampling Patterns

While producing satellite-like radiation fields is the main function of COSP-RTTOV, the implementation of satellite-like sampling patterns also enable new science applications.

#### 3.2.1 Separate evaluation of ascending and descending orbit branches

Sun-synchronous satellites make observations at two distinct times of day, allowing for investigations of both daytime and nighttime fields. Comparisons with ESMs, however, often only study the average field and lose the benefits of diurnal sampling. Figure 8 shows a comparison of both day- and nighttime AIRS radiances for a surface temperature channel for a mid-latitude region. This comparisons reveals that CESM2's wintertime cold bias occurs throughout the day, while compensating biases during the summer (too cold days, too warm nights) lead to good agreement in the average. Comparing averages over all orbits (e.g. Figure 5b) does not provide this insight. Satellite-like sampling in COSP-RTTOV enables these comparisons without saving high-frequency model output or running offline radiative transfer models.

#### 3.2.2 Quantification of Sampling Biases

The applications of COSP-RTTOV's diurnal sampling patterns can also be applied to standard COSP2 outputs. We demonstrate one application to precipitation frequency as observed by CloudSat, a spaceborne radar that flew from 2006-2023 measuring cloud structure and precipitation (Stephens et al., 2008). Figure 9a,b shows observed precipitation frequency from CloudSat for daytime and nighttime orbits (Haynes et al., 2009; Smalley et al., 2014). The diurnal contrast (Figure 9c) shows that precipitation frequency is generally highest over the ocean surface during the day and over the land surface during the night. Using satellite-like sampling COSP-RTTOV and COSP's CloudSat simulator (Kay et al., 2018), we generate comparable fields from CESM2 (Figure 9d-f). Qualitatively, CESM2 produces spatial patterns of precipitation frequency that closely resemble CloudSat. Differences between the simulated and observed fields (Figure 9g,h), however, show that CESM2 has too frequent precipitation during both day and nighttime orbits, leading to an overestimation of the diurnal contrast in precipitation frequency (Figure 9i). Over stratocumulus regions west of tropical continents, however, CESM2 completely misses the observed diurnal precipitation pattern. CloudSat shows greater nighttime precipitation frequency (Figure 9c), while CESM2 has near-

zero diurnal contrast (Figure 9f). We highlight that these comparisons require both consistent definitions of retrieved fields (provided by individual satellite simulator modules) as well as consistent definitions of their spatio-temporal sampling (provided by COSP-RTTOV). The large diurnal precipitation contrast also demonstrates that fair comparisons with the CloudSat record

following the 2011 battery anomaly (after which CloudSat only observed daytime scenes) require appropriate sampling patterns.

## 4 Conclusions

We developed a flexible and computationally efficient tool for simulating satellite-like radiation fields within Earth System Models. COSP-RTTOV is broadly applicable to satellite radiation fields in both clear and cloudy scenes. Furthermore, the

260 satellite-like radiation fields produced by COSP-RTTOV are consistent with instrument spectral response functions and orbit sampling, as well as the internal physics of the host model. The definition- and scale-aware comparisons enabled by COSP are thus broadly extended to studies using spectral infrared satellite observations. COSP-RTTOV emulates direct satellite observations that are tied to standards with known uncertainties. Evaluating ESMs against direct radiation observations is thus a strong test of their performance, and COSP-RTTOV is a single tool that enables such comparisons. Here, we have described

the design, validation, and potential uses of COSP-RTTOV. We demonstrate applications for short satellite missions, ESM evaluation, and climate change detection. Collectively, these examples demonstrate that COSP-RTTOV is a valuable tool for the modeling and observational communities. We welcome contributions of the broader community to further improvements to COSP-RTTOV.

. The current version of COSP-RTTOV is available at: https://github.com/jshaw35/COSPv2.0/tree/cesm2.2.0_rel_cosp_rttov under the MIT

licence. The exact version of the model used to produce the results used in this paper is archived on Zenodo (Shaw, 2025c), as are data (Shaw, 2025a) and scripts (Shaw, 2025b) to produce the plots for all the simulations presented in this paper.

## Appendix A

### A1

. J.K.S. and J.E.K. conceived the study. J.K.S. and D.J.S. developed the software. J.K.S. and D.S. ran climate model simulation experiments.

S. D-S. ran SARTA simulations and provided AIRS observational data. J.K.S. performed the analysis.

. The authors declare that no competing interests are present.

. J. K. S., D. P.S., and J. E. K were supported by the NASA PREFIRE mission Award 849K995. J. K. S. was additionally supported by NASA FINESST Grant 80NSSC22K1. Computing and data storage resources, including the Cheyenne (doi:10.5065/D6RX99HX) and Derecho (doi:10.5065/qx9a-pg09) supercomputers, were provided by the National Science Foundation's National Center for Atmospheric Research (NSF NCAR)'s Computational and Information Systems Laboratory. J. K. S. thanks the Polar Climate Working Group of the Community Earth System Model for computing resources. J. K. S. thanks James Hocking and the RTTOV development team for producing RTTOV coefficients for the NASA PREFIRE mission.

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

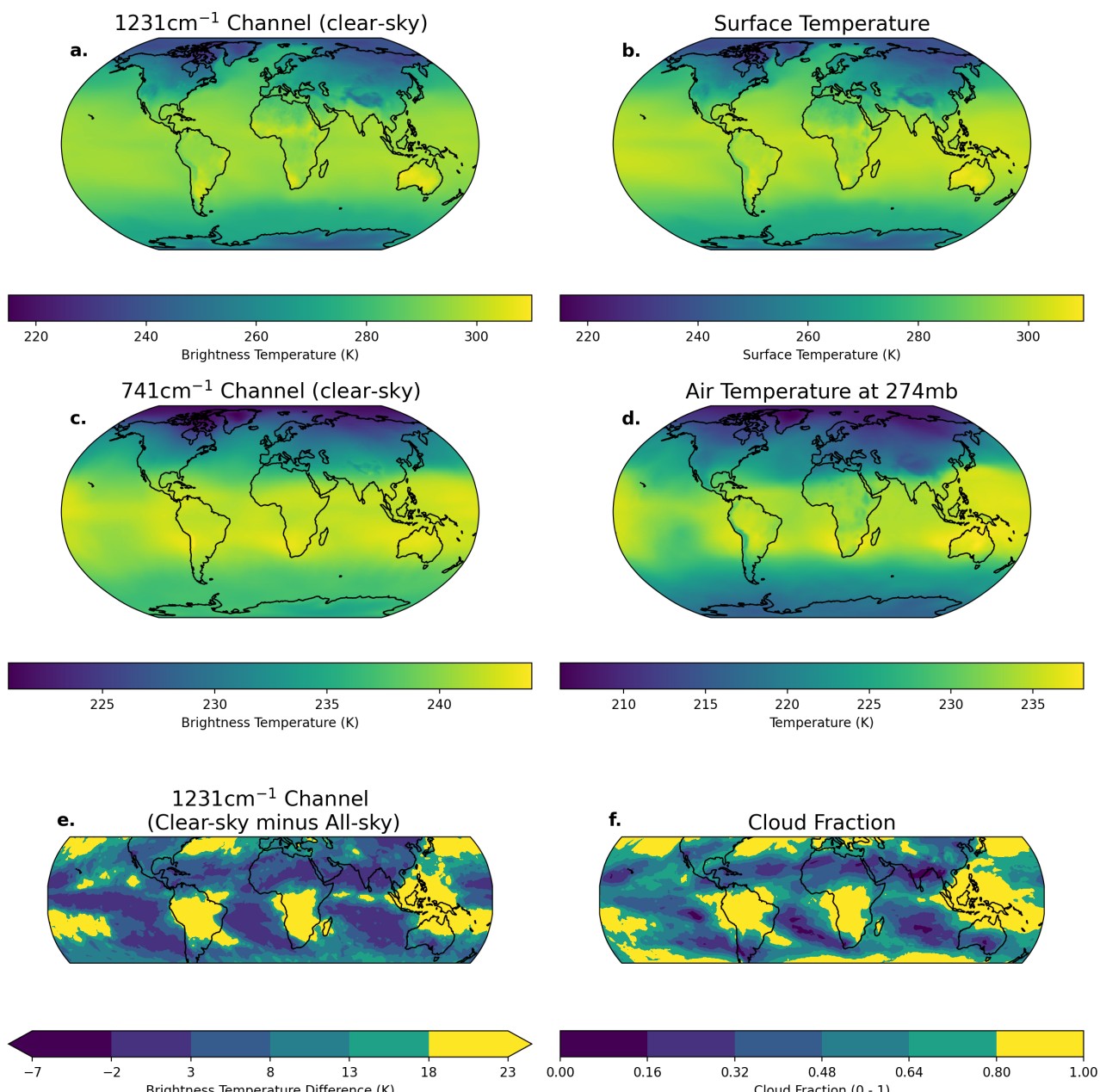

**Figure 4.** Comparison of COSP-RTTOV brightness temperatures and geophysical fields produced in a CESM2 historical simulation. a. Clear-sky brightness temperatures for a transparent AIRS window channel at $1231.33\text{cm}^{-1}$. b. Surface temperature. c. Clear-sky brightness temperatures for an AIRS channel sensitive to upper-troposphere temperature and CO2 at $740.97 \text{ cm}^{-1}$. d. Air temperature at 274mb. e. Difference between clear-sky and all-sky brightness temperatures in the $1231.33\text{cm}^{-1}$ AIRS window channel. f. Total cloud fraction. Panels e and f are restricted to 60S-60N where there is high thermal contrast between clouds and the surface. All plots represent average values for January 2000.

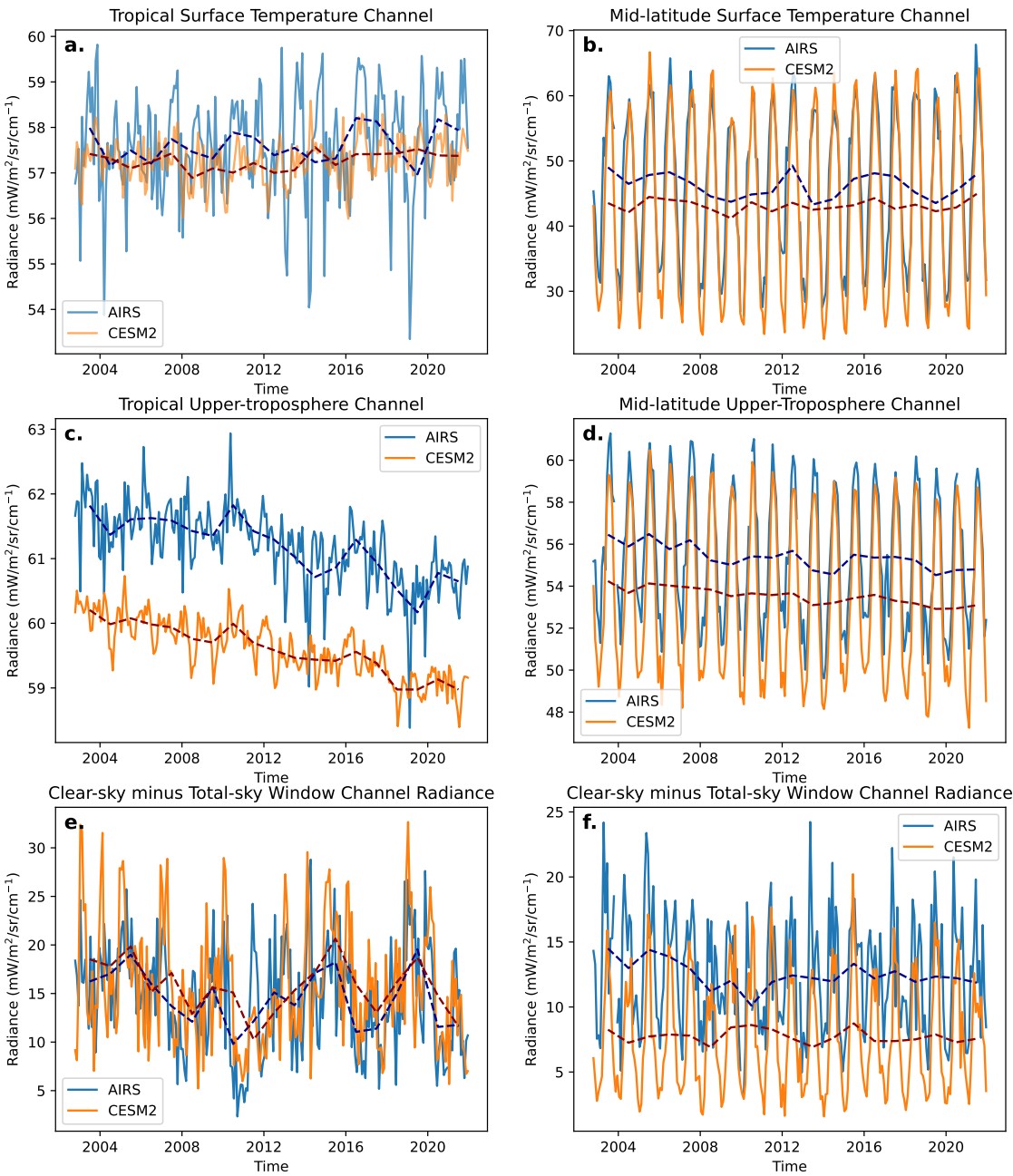

**Figure 5.** Evaluation of CESM2 against radiation time series using COSP-RTTOV. Rows show different spectral channels sampling surface temperatures, the upper-troposphere, and clouds as described in Figure 4. The left column shows results from a region in the Tropical Pacific (0-2.75°N, 160-165°E). The left column shows results over a land region in North America (44-46.75°N, W95-100°E).

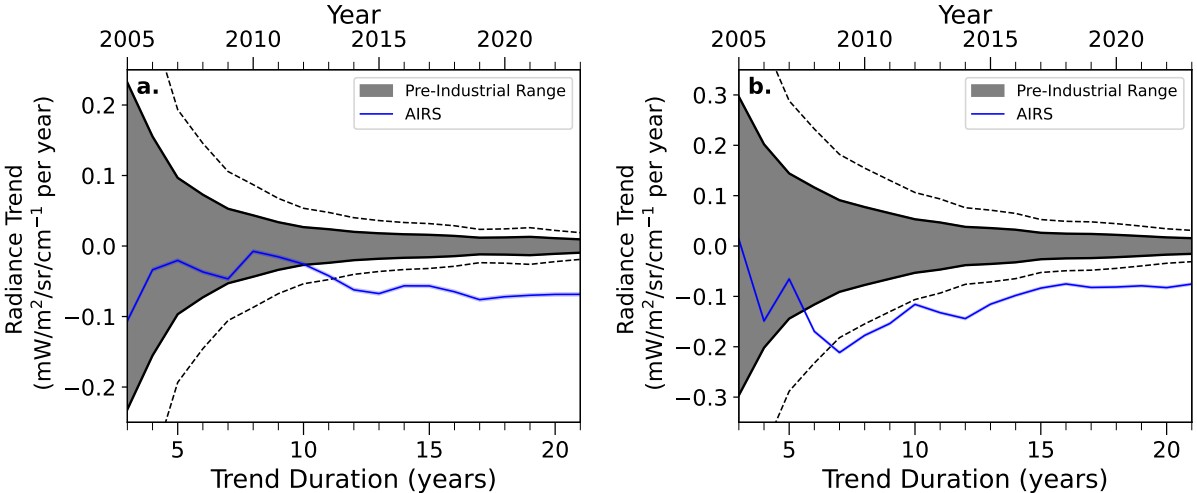

**Figure 6.** Time of Emergence of radiance trends for the AIRS 740.97 cm$^{-1}$ channel sampling the upper troposphere. Panel a and b show results from regions in the Tropical Pacific (0-2.75°N, 160-165°E) and North America (44-46.75°N, W95-100°E), respectively. Blue lines show trends in AIRS radiances. Trends begin in 2005 when three years data were available (2003-2005). Grey shaded regions span a 95% confidence interval on unforced trends calculated from a 199-year CESM2 pre-industrial control simulation following the methods of Shaw and Kay (2023) and Shaw and Lenssen (2024). Dotted grey lines double CESM2's estimate of internal variability in the case that regional variability is underestimated by CESM2. The AIRS record emerges from internal variability when it exits and remains outside of the envelope of internal variability.

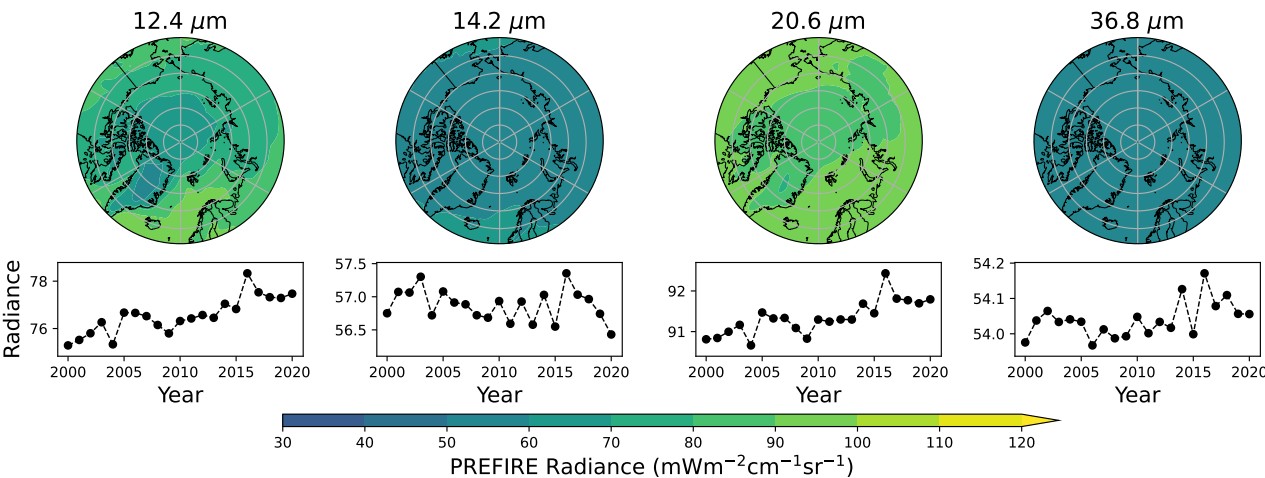

**Figure 7.** Top row: Polar maps of mean simulated radiances for PREFIRE channels at 12.4, 14.2, 20.6, and 36.8um over the 1979-2014 period. Bottom row: Time series of annual mean radiance values averaged over 60-90N show the combined effects of forced change and internal climate variability.

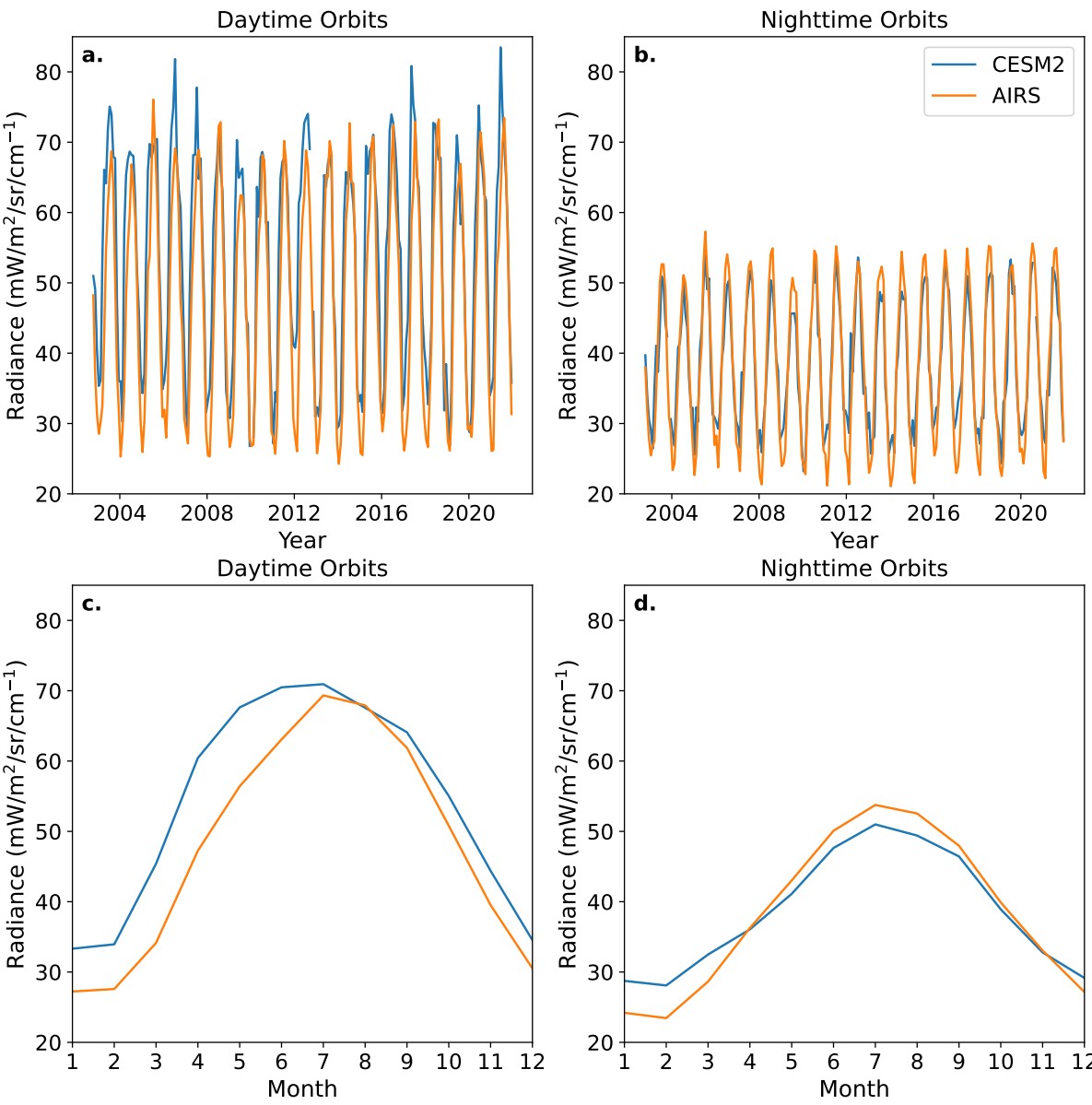

**Figure 8.** Comparison of AIRS and CESM2 for clear-sky radiances at $1231cm^{-1}$ for a land region in North America (44-46.75°N, W95-100°E) for ascending orbits (1:30pm local time) and descending orbits (1:30am local time). a. Monthly time series for ascending orbits. b. Monthly time series for descending orbits. c. Monthly climatology for ascending orbits. d. Monthly climatology for descending orbits.

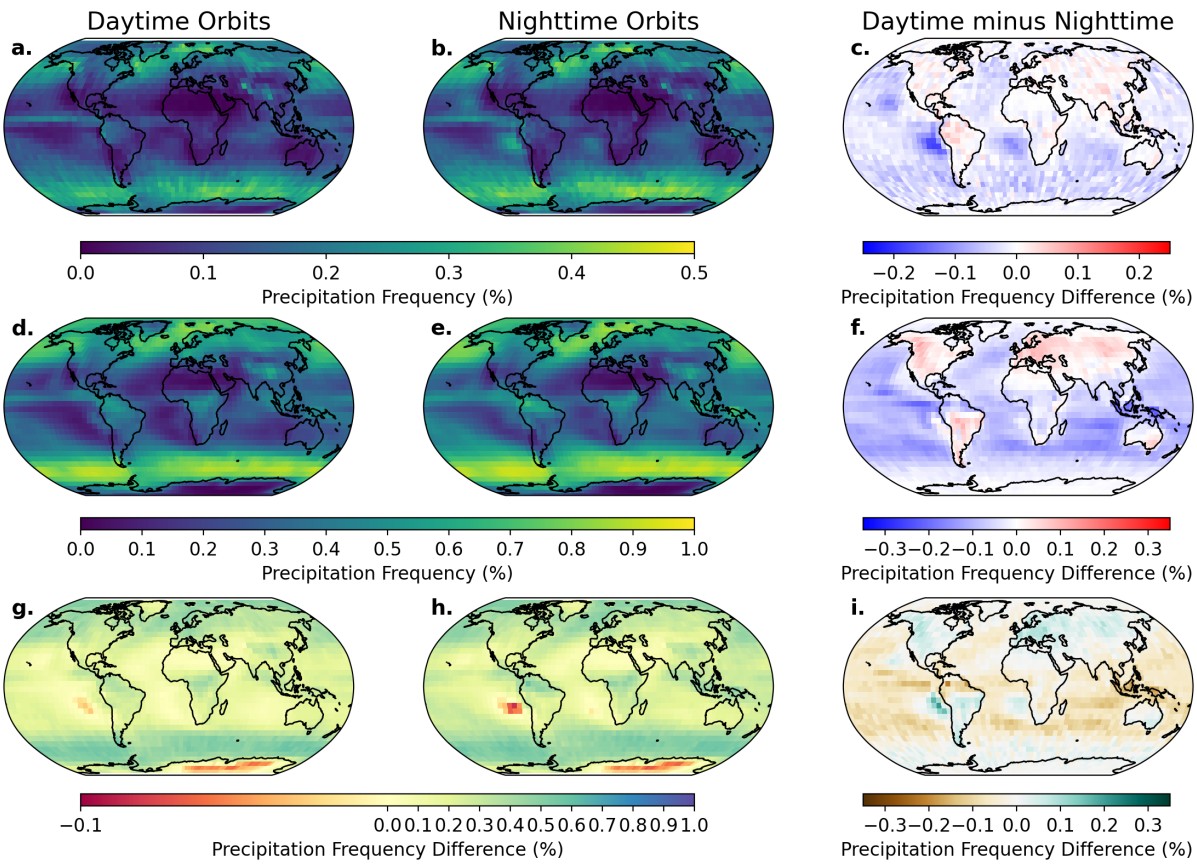

**Figure 9.** Comparison of day- and nighttime precipitation frequency from CloudSat and the COSP CloudSat simulator run in CESM2. a. Observed precipitation frequency for daytime (1:30pm) orbits. b. Observed precipitation frequency for nighttime (1:30am) orbits. c. Observed daytime minus nighttime precipitation frequency. d-f. Precipitation frequency as in panels a-c but from CESM2. g-i. Precipitation frequency as in panels a-c but CESM2 minus CloudSat observations. CloudSat observations and CESM2 output are averaged from June 2006 through May 2010. Note that colorbars in the top two rows have different ranges. CloudSat observations use Haynes et al. (2009) and Smalley et al. (2014). Modeled output uses the COSP CloudSat simulator (Kay et al., 2018).

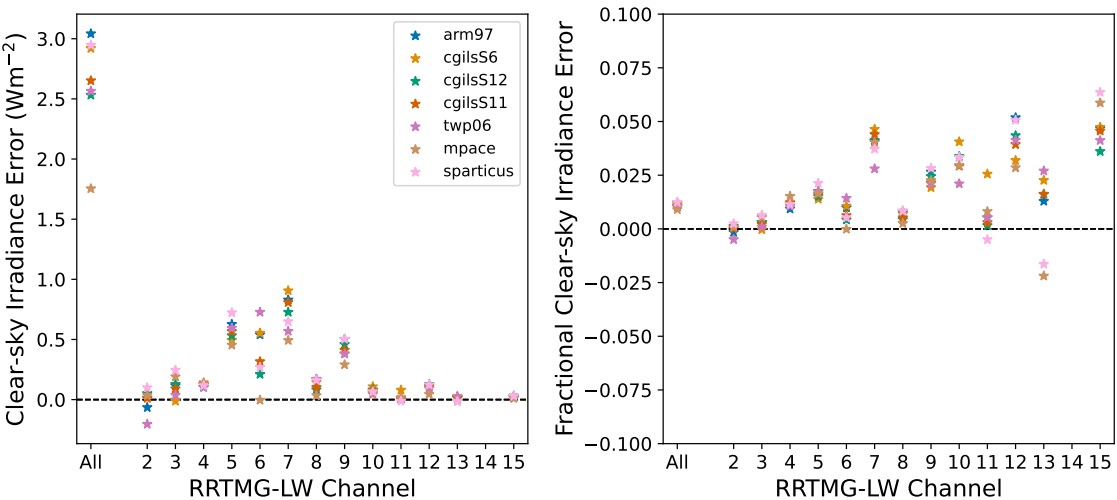

**Figure A1.** Comparison of clear-sky irradiances produced by COSP-RTTOV and CESM2 for 14 spectral bands in RRTMG-LW. a. Mean clear-sky irradiance error. b. Fractional clear-sky irradiance error. Temporal averages are taken before comparison between COSP-RTTOV and RRTMG-LW. Spectral boundaries in cm$^{-1}$ for RRTMG-LW bands 2-15 are 350, 500, 630, 700, 820, 980, 1080, 1180, 1390, 1480, 1800, 2080, 2250, 2380, 2600 (e.g. Band 3 spans 500-630 cm$^{-1}$). Band 14 samples the mesosphere above the CESM2 model top and is excluded from this analysis. RTTOV radiances are converted to irradiances using a 6-point gaussian quadrature with viewing zenith angles and weights following Stamnes et al. (2017).

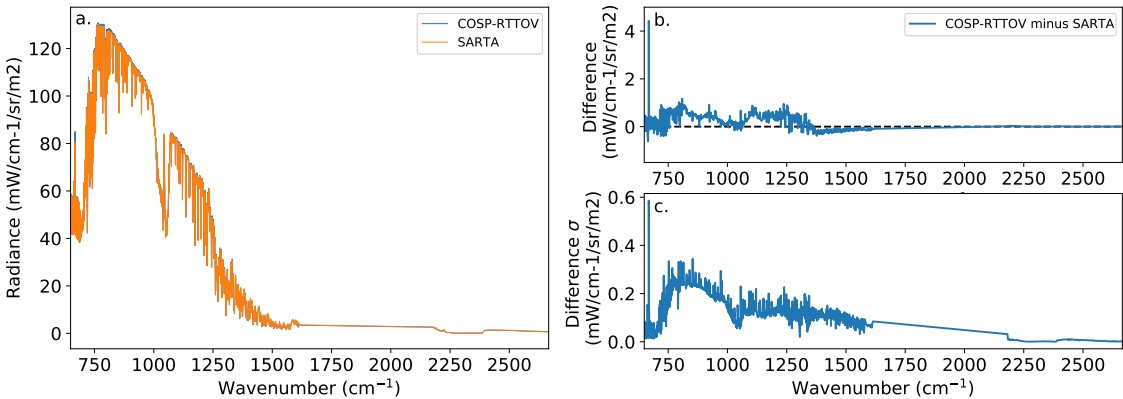

**Figure A2.** Comparison of radiances produced by COSP-RTTOV and SARTA for AIRS L1C channels. a. Simulated radiances across the AIRS spectral region ($3.7 - 15.4 \mu m$). b. Mean and c. Standard deviation of COSP-RTTOV radiance differences relative to SARTA. Radiances are computed for 333 atmospheric profiles taken from a single-column mid-latitude simulation of COSP-RTTOV (see section 2.1.4).