# Peer review of "COSP-RTTOV-1.0: Flexible radiation diagnostics to enable new science applications in model evaluation, climate change detection, and satellite mission design"

_EGUsphere, 2025_

## Author Comment (AC1)

**Author response to reviewer feedback on egusphere-2025-169, *COSP-RTTOV-1.0: Flexible radiation diagnostics to enable new science applications in model evaluation, climate change detection, and satellite mission design***

*Dear editor and reviewers,*

*Thank you for your valuable and constructive comments on our manuscript. Each reviewer raised useful and interesting questions that have improved the science and communication of our study. Our specific responses to each of the reviewers' comments are included below.*

*Best,*

*Jonah Shaw for the authors*

**Referee: 1**
**COMMENTS TO THE AUTHOR(S)**

This paper describes the development integration of the RTTOV radiative transfer code within COSP2, and presents a range of potential applications by using CESM2 as host model. The paper is well written and the content is adequate for publication in GMD. It provides a very valuable description of the new capabilities, including the satellite-like sampling patterns made available to all simulators within COSP. I particularly like Section 3, which covers a wide range of potential applications at the right level of detail for a paper documenting the new capabilities. Overall, I find the paper gives the reader an excellent overview of the potential of COSP-RTTOV for evaluation and climate studies without making the paper too long. I believe this paper is a useful addition to the scientific literature and deserves publication with only minor changes. Please see my specific comments below.

SPECIFIC COMMENTS

-L14-20. The first paragraph of the introduction would benefit from some references, for instance to the IPCC AR6.

We have added appropriate references to IPCC AR6, as well as references to Simpson et al., and Raghuraman et al.. (L15-22)

References:

Eyring, V., Gillett, N. P., Achuta Rao, K. M., Barimalala, R., Barreiro Parrillo, M., Bellouin, N., Cassou, C., Durack, P. J., Kosaka, Y., McGregor, S., Min, S., Morgenstern, O., and Sun, Y.: Human Influence on the Climate System, in: Climate Change 2021: The Physical Science Basis. Contribution of Working Group I to the Sixth Assessment Report of the Intergovernmental Panel on Climate Change, edited by Masson-Delmotte, V., Zhai, P., Pirani, A., Connors, S. L., Péan, C., Berger, S., Caud, N., Chen, Y., Goldfarb, L., Gomis, M. I., Huang, M., Leitzell, K., Lonnoy, E., Matthews, J. B. R., Maycock, T. K., Waterfield, T., Yelekçi, O., Yu, R., and Zhou, B., pp. 423–552, Cambridge University Press, Cambridge, United Kingdom and New York, NY, USA, https://doi.org/10.1017/9781009157896.005, 2021.

Lee, J.-Y., Marotzke, J., Bala, G., Cao, L., Corti, S., Dunne, J. P., Engelbrecht, F., Fischer, E., Fyfe, J. C., Jones, C., Maycock, A., Mutemi, J., Ndiaye, O., Panickal, S., and Zhou, T.: Future Global Climate: Scenario-Based Projections and Near-Term Information, in: Climate Change 2021: The Physical Science Basis. Contribution of Working Group I to the Sixth Assessment Report of the Intergovernmental Panel on Climate Change, edited by Masson-Delmotte, V., Zhai, P., Pirani, A., Connors, S. L., Péan, C., Berger, S., Caud, N., Chen, Y., Goldfarb, L., Gomis, M. I., Huang, M., Leitzell, K., Lonnoy, E., Matthews, J. B. R., Maycock, T. K., Waterfield, T., Yelekçi, O., Yu, R., and Zhou, B., chap. 4, Cambridge University Press, Cambridge, UK and New York, NY, USA, https://doi.org/10.1017/9781009157896.006, 2021.

Simpson, I. R., Shaw, T. A., Ceppi, P., Clement, A. C., Fischer, E., Grise, K. M., Pendergrass, A. G., Screen, J. A., Wills, R. C., Woollings, T., Blackport, R., Kang, J. M., and Po-Chedley, S.: Confronting Earth System Model trends with observations, Science Advances, 11, 8035,

https://doi.org/10.1126/SCIADV.ADT8035/ASSET/A8251B2C-2361-4850-92E0-9B77250E02AA/ASSETS/IMAGES/LARGE/SCIADV.ADT8035-F5.JPG, 2025.

Raghuraman, S. P., Paynter, D., and Ramaswamy, V.: Anthropogenic forcing and response yield observed positive trend in Earth's energy imbalance, Nature Communications 2021 12:1, 12, 1–10, https://doi.org/10.1038/s41467-021-24544-4, 2021.

-L45-47. I suggest adding references to previous work documenting uncertainties and limitations of simulators, e.g. https://doi.org/10.1175/JCLI-D-11-00267.1 and https://doi.org/10.1175/2010JCLI3517.1.

We have added the following sentence at lines L38-39.

While satellite simulators have their own limitations (e.g. Pincus et al., 2012; Mace et al., 2011), comparisons with observations are vastly improved over less-nuanced methods.

-Figure 4. I recommend adding coast lines to the figures to facilitate the interpretation of the maps.

We have made the suggested change to Figure 4.

-Figure 7. I suggest deleting the monthly means from the bottom row of plots (or moving them to a 3rd row of plots). This will allow the use of a better y-axis scale for the long-term averages.

We have made the suggested change to Figure 7 to show long-term trends.

-Figure 8. I believe the main point of this figure could be better made by plotting the average seasonal cycle, rather than the whole time series of monthly means.

We have made the suggested change to Figure 8.

**Referee: 2**
**COMMENTS TO THE AUTHOR(S)**

Implementation of RRTOV in the COSP is described along with tests to verify that it works correctly, mostly using single column model simulations for a range of cases. In addition to describing the implementation, the broader utility of RTTOV thermal radiances are shown using global climate model simulations with CAM. This includes the application of the RRTOV radiances for model evaluation, OSSEs and showing the effect of sampling. The paper is well organized and provides sufficient information about the RTTOV implementation in COSP and relevant applications.

Comments

Line 28 and 29: "only measure spectrally-resolved radiation field", true for passive instruments but there are active instruments that measure returns from a known source signal.

We have replaced "Satellite observations" with "Passive satellite observations" to clarify our statement.

Line 43: Just a point that instruments on satellites do not directly observe fields like radiances, e.g., the L0 data needs to be processed into a L1 quantity like radiances which have uncertainties.

We have made the following changes at lines L45-46:

Radiation fields, on the other hand, are derived directly from L0 observations and can be accurately calculated in ESMs using radiative transfer models.

Line 119: The CESM simulation using pre-industrial boundary conditions is 199 years long?

Yes. The boundary conditions for this simulation come from years 500-699 but output is not produced until February of year 500 (due to boundary condition application in the middle of January of year 500), so only 199 complete years are produced. While an even number of years would be ideal, we are confident this length does not influence our results.

Line 125: Is SARTA faster than RTTOV? While RTTOV offers flexibility, if one was interested in evaluation relative to AIRS there could be value in using SARTA instead of RTTOV given the computational cost (Table 3).

Because we have not installed SARTA into COSP, we do not have an estimate of the comparison between the cost of in-line simulations, though would estimate a similar computational cost. Installing both models into COSP and comparing their performance is beyond the scope of this project, so we have not modified the manuscript.

Line 139: Are there options in RTTOV to use different cloud optics over cloud vertical overlap? While it might not be possible to match exactly what is used in the host model, there would be value if assumptions could be similar as possible. E.g., using the same cloud vertical overlap assumption.

RTTOV only allows for two cloud overlap assumptions, the default recommended maximum/random overlap and a simplified cloud overlap approach with one clear and one cloudy column. We have updated the following sentence in section 2.1.4 to describe these specifics and options for cloud optics, all of which may be modified within COSP-RTTOV. For detailed questions, we refer readers to the RTTOV documentation.

Changes at lines L142-144:

*RTTOV, however, has its own schemes for cloud optics and cloud overlap assumptions (how clouds at different vertical levels are distributed at sub-grid scales). Specifically, users may select cloud optics from multiple parameterizations, and choose between maximum/random and simplified two-column cloud overlap schemes (Saunders et al., 2018).*

Section 3.1.1: As noted in this section, different AIRS channels provide different information about the simulation. With respect to the computational time, the end user may decide to only simulate particular channels. Is this easily configured in COSP-RTTOV?

Yes, simulating a subset of channels is easily configurable. This is described in Figure 1 and section 2.1.1. To clarify that a subset of channels may be produced, we have made the following changes to lines L101-102 in Section 2.1.1:

*To determine what radiation fields are produced, users specify which subset of spectral channels and output fields should be produced for each instrument being simulated by RTTOV.*

We have also specified that only a subset of AIRS channels are simulated in our global ESM experiments. The following line has been added at lines L117-118 in Section 2.1.2:

*In both experiments, only a subset of AIRS channels are simulated to limit computational cost and the volume of data produced.*